# Drugit: crowd-sourcing molecular design of non-peptidic VHL binders

Building on the role of human intuition in small molecule drug design, we explored whether crowdsourcing could recruit citizen scientists to this task while in parallel building awareness for this scientific process. Here, we introduce Drugit (https://drugit.org), the small molecule design mode of the online citizen science game Foldit. We demonstrate its utility by identifying distinct binders to the von Hippel Lindau E3 ligase. Several thousand molecules were suggested by players in a series of ten puzzle rounds. The proposed molecules were further evaluated in silico and manually by an expert panel. Selected candidates were synthesized and tested. One of these molecules shows dose-dependent shift perturbations in protein-observed NMR experiments. The co-crystal structure in complex with the E3 ligase reveals that the observed binding mode matches the player's original idea. The completion of one full design cycle is a proof of concept for the Drugit approach and highlights the potential of involving citizen scientists in early drug discovery.

Despite continued efforts to use rational approaches to automate and accelerate drug development, creativity and human intuition are (still) required. Currently, this role is fulfilled by highly trained people in the pharmaceutical industry or academia. However, crowdsourcing games such as Phylo[1], GalaxyZoo[2], and Eterna[3], together with open science platforms such as SGC's open science[4,5] or Boehringer Ingelheim's opnMe.com[6], allow citizen scientists to contribute to open research questions, adding value to open innovation. One of the earliest citizen science games, Foldit is a molecular biology modeling game which has previously been shown to leverage the creative potential of citizen scientists to predict and design protein structures[7–10]. This is made possible by combining protein modeling tools with scoring and incentivization. The Foldit interface is tailored to empower players with limited prior knowledge to solve molecular design problems, and through it, members of the public can suggest protein structure modifications which can be experimentally verified.

The Von Hippel-Lindau (VHL) E3 ligase is a component of the CUL2-RBX1-ElongingB-ElonginC-VHL cullin-RING ubiquitin ligase complex, and its natural substrates are hypoxia-inducible factor (HIF) proteins hydroxylated at a conserved proline amino acid by prolyl-hydroxylase domain proteins[11]. It is one of the commonly used E3 ligases for proteolysis targeting chimeras (PROTACs), a recent drug modality which holds promise for future therapeutic strategies[12]. PROTACs are bi-functional in nature, linking a moiety that is specific for a target protein with a moiety that is E3 ligase-specific. A single E3 ligase moiety can be combined with a range of protein-specific moieties to create PROTACs which prime different target proteins for degradation[13]. VHL poses an excellent test case as example compounds that bind tightly to VHL exist and PROTACs have been derived[14,15]. Currently, the chemical diversity of known VHL-small molecules is limited. While extensive optimization has been performed on molecules originally introduced by Crews and Ciulli[16], including progress in de-peptidizing parts of the molecule to reduce polarity, the common hydroxyproline core crucial for binding affinity contributes significant polar surface area[17]. Current molecules are thus limited in their potential to yield orally available PROTACs due to their poor pharmacokinetic properties, particularly their high efflux as well as poor permeability and stability[18].

Here we describe the engagement of a larger non-expert community in the drug design process. We describe an extension of the Foldit interface with a small molecule design tool ("Drugit", https://drugit.org) based on the RosettaLigand scoring and sampling algorithm[19–21]. Using VHL binder design as a test case, we address the multifactorial needs of drug design by tuning the objectives of the

✉ e-mail: jens@meilerlab.org; jark.boettcher@boehringer-ingelheim.com; rocco.moretti@vanderbilt.edu

puzzles not only for creating molecules with high affinity to VHL, but also including penalties and/or bonuses to control topological polar surface area (TPSA), hydrogen bond donor count, and cLogP.

## Results

### Drugit interface

Foldit design projects proceed as a series of puzzles, which are individual competitions lasting about a week. Each puzzle in a series can have a different starting molecule, protein context, scoring details, additional objectives, and tool configuration. Players use the tools available to modify the molecular system and compete to have the best score in each puzzle[7,9]. In small molecule design puzzles, the player is presented with the 3D structure of a starting small molecule docked, with the goal to optimize its structure and placement within the desired target protein binding pocket (Fig. 1). Via the small molecule design panel (Fig. 1a, b) the player can add or delete atoms and bonds, change element identity, or change the bond order. To accelerate the design process and encourage drug-like properties, a set of pre-defined fragments and functional groups frequently observed in drug-like molecules is provided. All changes to the molecule are instantaneously checked for basic chemical feasibility prior to acceptance, and players are informed of grossly unphysical molecules via pop-up messages. Simple rules for protonation state and hydrogen placement are

automatically applied to ensure that the small molecule is modeled in the correct charge state for physiological conditions. The edited molecule is then aligned with the existing molecule within the current binding pose, and the new chemical identity is inserted into the modeled protein.

### Geometry optimization

The overall geometry of the complex, including small molecule placement as well as internal coordinates of protein and small molecule, can be optimized in the Drugit framework (Fig. 1c). However, for this work, it was decided to hold protein atoms fixed, and only allow the small molecule to move. This more closely matches the prevalent molecule design approach in early-stage drug discovery and should focus players' efforts on compounds which match the binding mode of the starting point. More diverse compounds might be found in future experiments using proteins with more flexibility.

The primary optimization tool is gradient descent minimization within the Rosetta energy function (wiggle). To help correct for unfavorable geometries, small molecule design puzzles use a dual space minimization which alternates between Cartesian and internal coordinate degrees of freedom[22]. Ideal geometries for small molecules are enforced by the cart_bonded geometric deviation penalty[23], with ideal geometries derived from the Merck molecular force field (MMFF)

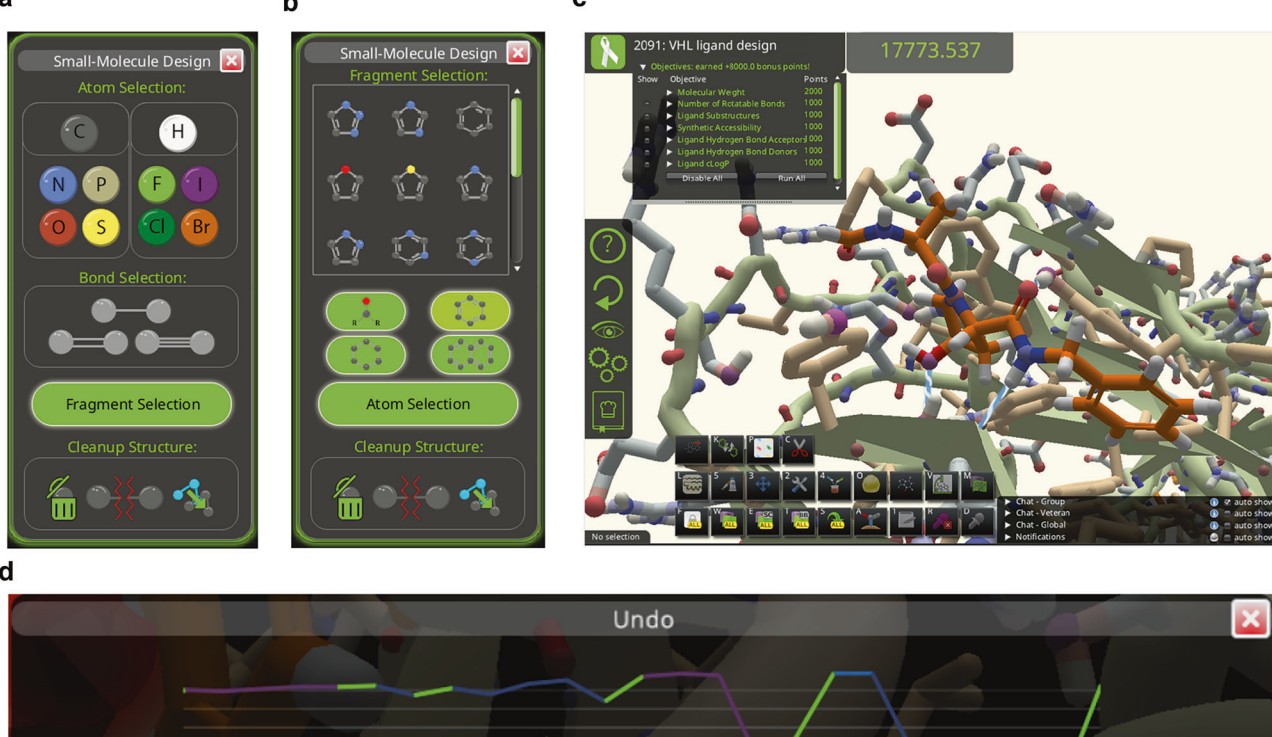

**Fig. 1 | The Drugit user interface. a** Players can use the atomistic design panel to add and delete atoms and bonds. **b** The fragment tool allows players to quickly insert functional groups and ring systems. **c** Once compounds have been designed, players can optimize their placement with an assortment of tools. Objectives are designed to guide players by increasing the overall compound score. **d** The undo utility function tracks the player's progress as they build and manipulate their

designed compounds. This function offers a graphical representation of the changes they have made. Each change is denoted by a color associated with the respective change. In the Figure, changes made by the builder are represented in blue, while other changes such as minimization or wiggle are shown in green. Players can use this tool to go back to a state that scored highly and build in different directions.

values[24]. An alternative minimization under MMFF directly is also available.

Conformational sampling of the small molecule can be carried out using the shake tool. This currently has limited usefulness, due to the need to generate conformers for the small molecules on the fly. While multi-threading allows some calculation of conformers in the background, due to the speed of the ETKDG conformer generation[25] code currently being used, only a small set of conformers are sampled.

A final optimization approach is the manual pull tool, which allows players to grab portions of the molecule and drag them from their current location to a new location in space. As the tool is intended to be used also by non-expert users and a significant proportion of trial and error in small molecule design is foreseen, an undo function is implemented. This function tracks multiple saved states, allowing the player to step backward in their building timeline to either previously created structures of interest, or structures that scored higher than their current iteration (Fig. 1d).

### In-client geometry evaluation

The primary evaluation of binding is the Rosetta energy of the protein and small molecule system. This includes the cart_bonded term which penalizes small molecule with strained internal geometries. The protein/small molecule interaction energy can be upweighted to increase the importance of a good binding energy.

As binding energy is not the only consideration for drug candidates, Drugit puzzles make extensive use of objectives, i.e., additional bonuses and penalties which are added to the player's scores for system properties such as molecular weight, topological polar surface area (TPSA)[26], cLogP[27], number of hydrogen bond donors or acceptors, synthetic accessibility[28], or the presence of undesired substructures. These metrics are currently mostly calculated through the RDKit library (https://www.rdkit.org/). The importance and thresholds of each of these settings can be adjusted on a per-puzzle basis. In addition, most objectives come with a visualization which can highlight those atoms or regions, causing a suboptimal score for a particular objective.

### VHL puzzle series

The Drugit VHL Puzzle series is a set of ten puzzles released over consecutive weeks from Oct 20, 2021 to Jan 12, 2022. The puzzles were based on the structure of the Von Hippel-Lindau (VHL) disease tumor suppressor protein in complex with ligand 10, a previously reported binder (PDB ID: 5NVX, Fig. 2a), named here as reference molecule 1 (Fig. 2). Like all reported potent small molecules, reference molecule 1 possesses a central hydroxyproline core motif, mimicking the natural substrate and amide-linked substituents embedded in the peptide binding groove. Players were instructed to both vary the core and find replacements as well as reduce compound polarity, provided that cellular efflux has been identified as a potential limitation of VHL-based PROTAC therapeutics. Particularly, the TPSA was used as a proxy for efflux potential. The ten rounds of the puzzle series maintained the protein structure and the overall goals, but varied in the starting small molecules, the objectives present, and the relative weights of these objectives (Supplementary Tables 1, 2). The preliminary results from each round were examined by medicinal chemists, and set limits were adjusted according to in-house criteria and expert opinion on the quality of observed compounds. Objectives were added or reweighted to iteratively improve the compounds observed. For example, in the first round, no penalty for the lipophilicity of the compounds was included, leading to excessively hydrophobic compounds. The addition of an objective which penalized compounds with high cLogP immediately resulted in compounds in a more desirable range in the next puzzle (Supplementary Fig. 1). Crystallographic waters and additional bonuses for hydrogen bonds to those waters were also added to the starting structure to improve the quality of the submitted molecules.

Over the course of the puzzle series, 333 Foldit players loaded the puzzles, and 160 contributed at least one novel compound. The number of distinct compounds submitted per player roughly follows an exponential decay relationship, with the number of compounds per player dropping by roughly one-half for every 17 places in the prolificity ranking (Supplementary Fig. 2).

### Post-competition filtering

The Foldit client captures not only the best-scoring compound for each player but also allows players to upload compounds which are interesting for later scientific evaluation. Further, the client also takes regular snapshots of each player's progress. All structures submitted to the server were assembled for analysis and deduplicated on chemical identity. The best-scoring geometry for each compound was taken as representative. For compounds present in multiple rounds, structures from later rounds were selected.

During VHL post-game triaging, ca. 6500 compounds were reduced to 19 using a combination of computational chemistry tools and medicinal chemistry judgment. First, molecules with any atom more than 10 Å away from any starting molecule binding site atom were removed. Second, Pipeline Pilot was used for automatically filtering out chemically unreasonable molecules, via element composition or property filters (e.g., TPSA, clogP, efflux and permeability predictions, and SMARTS custom filters [see SI for details]). Figure 2b shows the 1073 remaining player suggestions within the VHL binding site, while Fig. 2c compares the chemical space covered by player suggestions vs. the known VHL binders used as spike molecules during property filtering. Interestingly, the player-designed molecules (black dots) cover a wider chemical space, as defined by the two main principal component axes than the known VHL binders (black dots with blue circles). Third, the 1073 molecules were triaged in SeeSAR (SeeSAR version 12.1.0) as well as Flare (Cresset Discovery Services), retaining >80 compounds with a total TPSA below 100 Å$^2$ and >20 compounds with a TPSA between 100 and 120 Å$^2$, each featuring reasonable conformations and no intra- or intermolecular clashes. Additional compounds with suboptimal but still acceptable geometries and TPSAs below 100 Å$^2$ were also kept. Some compounds from the original game output were rescued visually from the original players' designs and >90 compounds were slightly modified based on medicinal chemistry knowledge, e.g., stabilizing torsional profiles by appropriate substituents and tuning ligand-protein complementarity in anticipation of improved affinity. Additionally, proposed pharmacophores with limited synthetic precedents were, if feasible, modified to improve synthetic accessibility, but the intended molecular interactions were kept.

Next, for a total of >260 compounds, MDCK efflux, Caco2 efflux, and apical to basal (AB) Caco2 permeability predictions were carried out using in-house implementations of various machine learning models embedded in Pipeline Pilot (see SI for details). Compounds with acceptable predictions for efflux and permeability were then redocked and their overall torsion quality was assessed (see SI for details). Based on visual inspection, computational chemists then agreed on a set of molecules predicted to make favorable interactions with the protein binding pocket to be taken further. In parallel, medicinal chemists inspected all docking poses visually and modified certain ideas from Drugit players slightly in anticipation of better physicochemical properties.

In a final computational assessment, >50 compounds were submitted to absolute binding free energy calculations (ABFE), including the known VHL binder reference molecule 1 as control. The binding affinity of all these compounds was then calculated by two different methods: FEP+[29,30] as well as the ABFE approach by Biggin and collaborators[31].

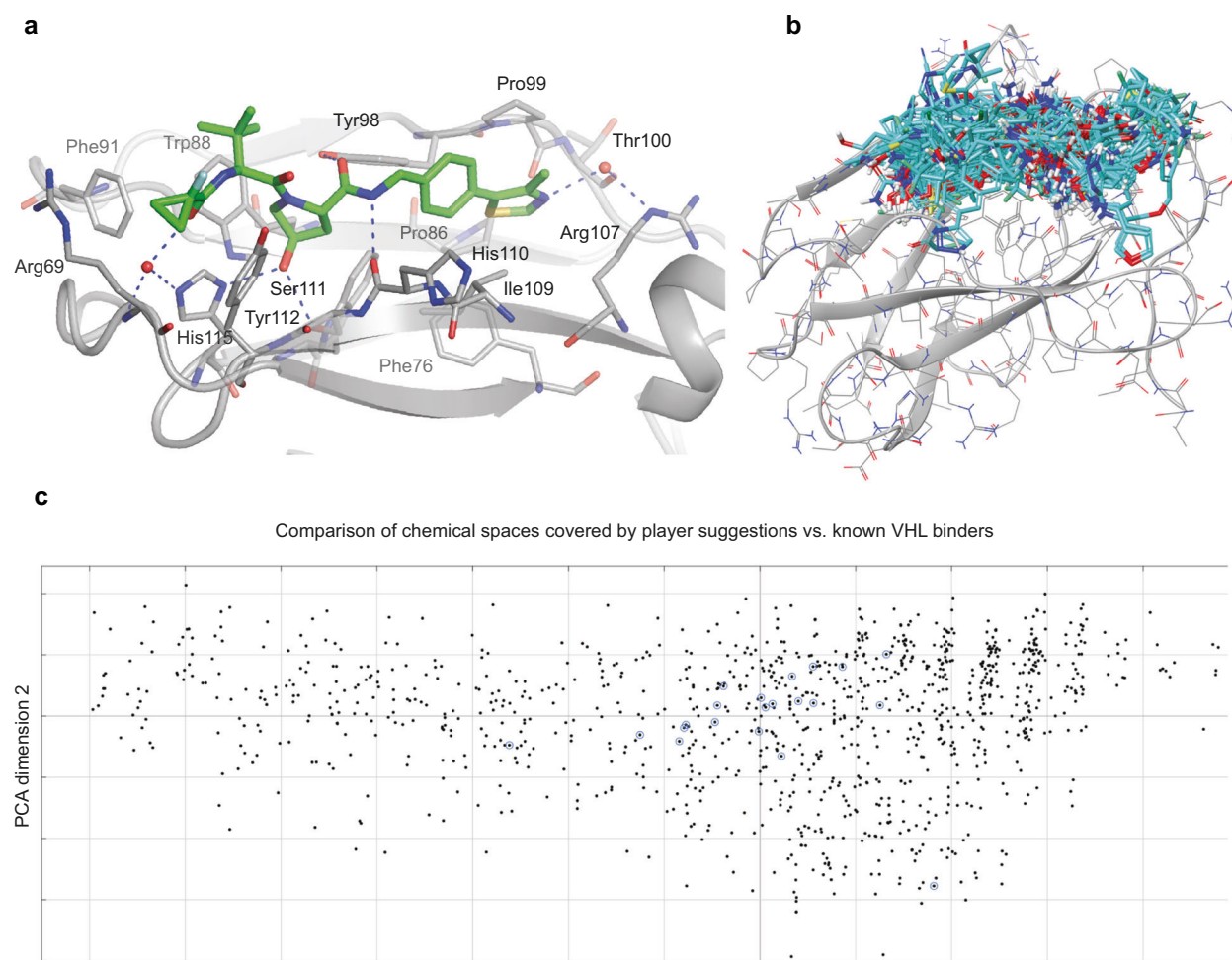

**Fig. 2 | Puzzle starting point and player suggestions. a** Binding mode of the puzzle starting point reference molecule 1, as observed in the previously reported crystal structure (PDB code: 5NVX; VHL in gray). The reference molecule 1 is shown as sticks and color-coded by atom type (with green carbons). **b** 1073 player suggestions in the VHL binding site: The suggestions are displayed as sticks and color-coded by atom type (with cyan carbons). **c** Comparison of the chemical space covered by player suggestions (black dots) vs. known VHL binders used as spike molecules (black dots with blue circles) during property filtering. The chemical space projection is based on the calculation of standard RDkit descriptors, a principal component analysis (PCA), and the reduction of the multidimensional descriptor space to the two dimensions explaining most of the variance. RDkit descriptor calculations and a PCA were carried out with default settings in MOE 2022.02 (Molecular Operating Environment, Chemical Computing Group), including a normalization of descriptors.

Lastly, based on the following criteria, a shortlist of 19 compounds was selected for assessment by experts: (i) compounds must be easily synthesizable within resource constraints; (ii) compounds must be likely stable, e.g., with functionality represented in marketed compounds; (iii) only one representative per core (i.e., binding either to Ser111 or His115), with left-hand side (LHS) and right-hand side (RHS) functional groups allowed; (iv) some chemotypes were excluded based on free energy calculations. After the rank-ordering of the compounds by experts, the selected molecules proceeded to synthesis.

**Synthesis and in vitro analysis of designed compounds**

The prioritized molecules were evaluated for synthesizability and, in some cases, needed minor adaptation. To increase redundancy and likelihood of success, precursors and synthetic intermediates were also submitted for testing (compounds submitted for testing in Supplementary Table 3; FEP+ predictions in Supplementary Table 4; calculated properties of synthesized compounds in Supplementary Table 5). Synthesis routes and data for 43 molecules generated in this study are described in the Supplementary Methods. Synthesized

compounds were submitted to a fluorescence resonance energy transfer (TR-FRET) assay using a Cy5-labeled VHL Tracer analog[32], an NMR-based solubility assay, and a $^{19}$F NMR displacement assay[33] (Supplementary Table 6). The latter is a highly sensitive technique to confirm that the compounds bind to the protein pocket of interest. Here, a well characterized $^{19}$F-containing reporter molecule was used at 50 μM concentration in presence of 2 μM VCB (VHL-ElonginC-ElonginB) complex. The addition of the compounds to be tested at 500 μM concentration led to different degrees of displacement of the reporter probe from the binding site, and, therefore, to the reappearance of the $^{19}$F NMR signal in the spectrum (Fig. 3a). The diastereomeric mixture of compound **1** was selected for further profiling as it showed a competitive behavior in the TR-FRET assay (IC$_{50}$ = 264 ± 30 μM), excellent solubility (>500 μM), and a dose-dependent displacement of the $^{19}$F NMR probe (52% recovery). **1** was submitted to protein-observed NMR experiments: Protein labeled selectively, with $^{13}$C-methyl groups in the residues Ile, Val, Leu, and Met, was used to obtain a K$_D$ = 182 ± 76 μM by fitting the dose-dependent shifts (Fig. 3b–d). The shift pattern induced by **1** is very similar to the ones induced by published VHL ligand VH298 and an initial starting point in VHL ligand discovery (named reference

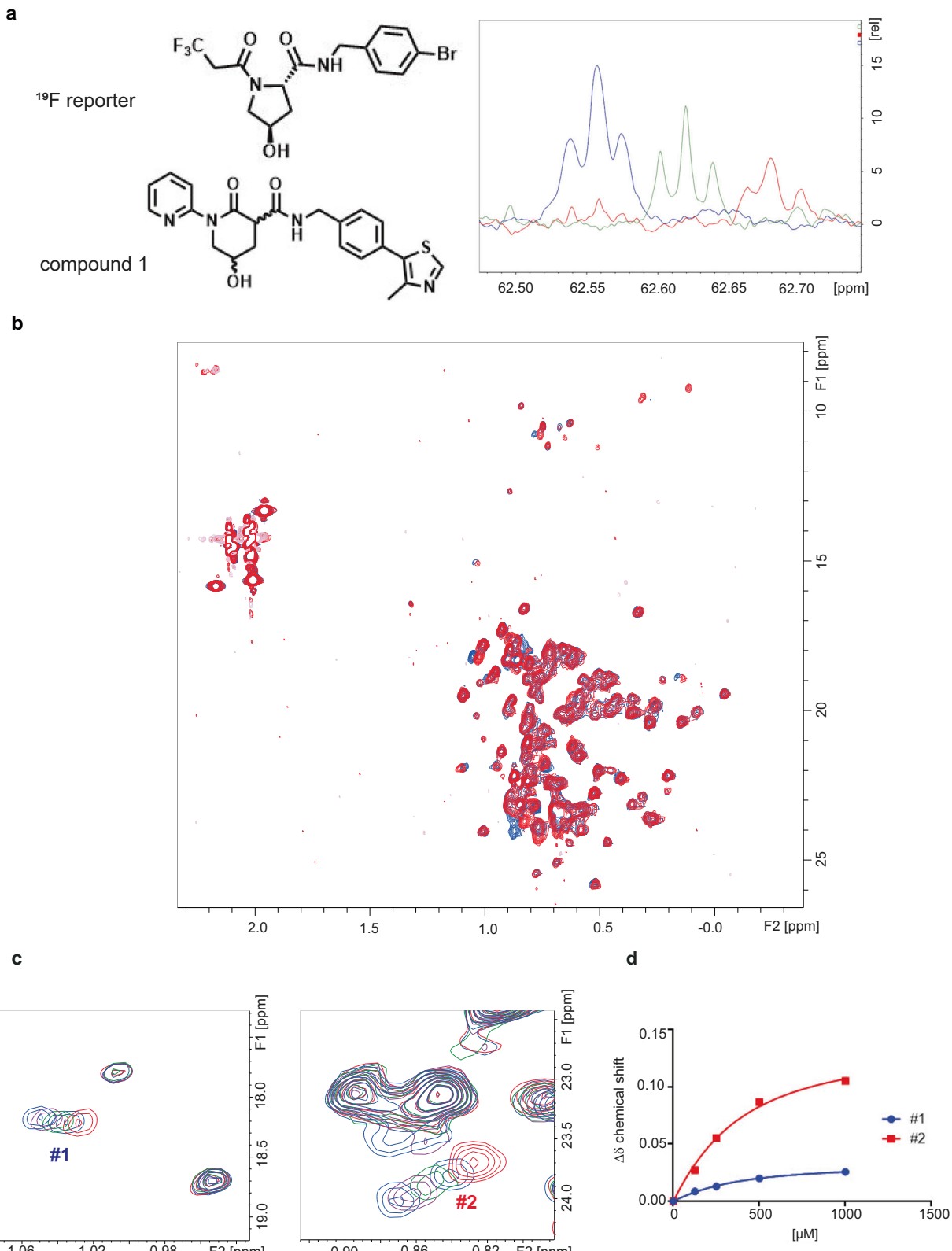

**Fig. 3 | Biophysical triaging and hit characterization. a** Structures of the [19]F reporter molecule used to detect binding and of **1**. [19]F NMR spectrum of 50 μM of the reporter in the presence (red) and absence (blue) of 2 μM VCB complex. Addition of 500 μM of **1** leads to reappearance of reporter signal due to its displacement from the binding site (green). **b** Methyl group region of specifically [13]C labeled VCB complex in the absence (red) and presence (blue) of 500 μM of **1**. **c** Expanded regions of signals #1 and #2 monitored during titration of 125, 250, 500, and 1000 μM of **1**. **d** $K_D$ values obtained by fitting the dose-dependent chemical shifts of peaks #1 and #2.

molecule 2 in this study) (Table 1 and Supplementary Fig. 3). Based on these encouraging results, the diastereomeric mixture of **1** was co-crystallized in complex with VCB. A crystal structure of the compound bound to VHL was obtained at a resolution of 1.98 Å. The sidechain and backbone conformations of ligand-contacting residues are not appreciably different from that of the reference molecule 1 bound structure (PDB ID: 5NVX). The well-resolved electron density allowed the identification of the eutomer (Fig. 4a, Supplementary Table 7, and Supplementary Fig. 4a, b). The hydroxy-piperidinone motif of **1** occupies the hydroxyproline recognition site of VHL, forming con-served hydrogen bonds to Ser111 and His115. While the RHS motif is conserved compared to reference molecule 1, ring expansion and amide inversion leads to an altered exit vector towards the LHS. Fur-ther optimization of **1** to engage in pi-pi interactions with Trp88 and Phe91 from a pyridine substituent can be envisioned (Fig. 4a, b). Interestingly, the eutomer observed in the co-crystal structures exhi-bits the ($R,R$) configuration, not matching the stereochemistry of the original player-designed compound with the ($R,S$) configuration, but does maintain the key pharmacophores (Fig. 4c, d). To allow for a direct comparison, absolute binding free energies for reference molecule 1, the bound diastereomer of **1**, and the player-designed molecule were calculated. The predictions for **1** and the player-designed molecule are within 1 kcal/mol and, therefore, in the same range, confirming that the key pharmacophores of the player's design are conserved (Supplementary Table 8).

### Table 1 | Comparison of compound properties

| | 1 (mixture of 4 diastereomers) | Reference molecule 1 | Reference molecule 2 |
|---|---|---|---|
| **Affinity/potency** | | | |
| NMR $K_D$ [µM] | 182 ± 76 | <10 | 29 ± 1 |
| TR-FRET $IC_{50}$ [µM] | 264 ± 30 | 0.090* | 132 ± 38 |
| **Basic molecular descriptors** | | | |
| cLogP | 1.21 | 2.68 | 0.83 |
| Molecular weight | 422.5 | 516.6 | 377.8 |
| H-bond acceptors | 5 | 5 | 4 |
| H-bond donors | 2 | 3 | 2 |
| TPSA (Å$^2$) | 95 | 111 | 95 |
| Rotatable bonds | 3 | 6 | 5 |
| Heavy atoms | 30 | 36 | 26 |
| $Fsp^3$ | 0.27 | 0.54 | 0.39 |
| **In vitro ADME properties** | | | |
| Aqueous solubility (pH 6.8) [mg/mL] | >0.098 | >0.115 | >0.088 |
| Caco2 permeability $P_{app, a-b}$ [cm/s] / efflux ratio | 23.0 × 10$^{-6}$/1.4 | 2.9 × 10$^{-6}$/15.5 | n.d. due to low recovery/4.8 |

TR-FRET data were reported as a mean value of seven independent experiments ± s.d for 1 and reference molecule 2. NMR $K_D$ values are calculated from two independent titrations and the averaged results of peaks #1 and #2 from Fig. 3d. *FP assay as reported previously.

## Who designed the winning molecule

The 19 molecules selected for evaluation came from nine different players with a range of Foldit experience. Compound selection was performed independently, and player identity was blinded. However, six of the selected compounds came from an experienced medicinal chemist on the evaluation team (C.A.P.S.), who had played the puzzles in his free time, but none of these compounds showed any binding. The compound that showed binding was based on a round 6 design from player Nicm25, who had previously played Foldit from 2011 to 2015, and resumed in early 2020, and who has no formal medicinal chemistry experience (personal communication). While Foldit has the ability for group play (players share structures among an organized group of players), the binding compound was developed indepen-dently (soloist).

There was no consistent pattern as to which puzzle round pro-duced the selected compounds. These compounds were generally not the best compounds of the round, nor were they appreciably related to the best-scoring designs for the respective player for that round. Neither the parent compounds nor the compound which was altered for synthesis showed an appreciable difference in Rosetta binding energy or docking quality (Supplementary Fig. 5).

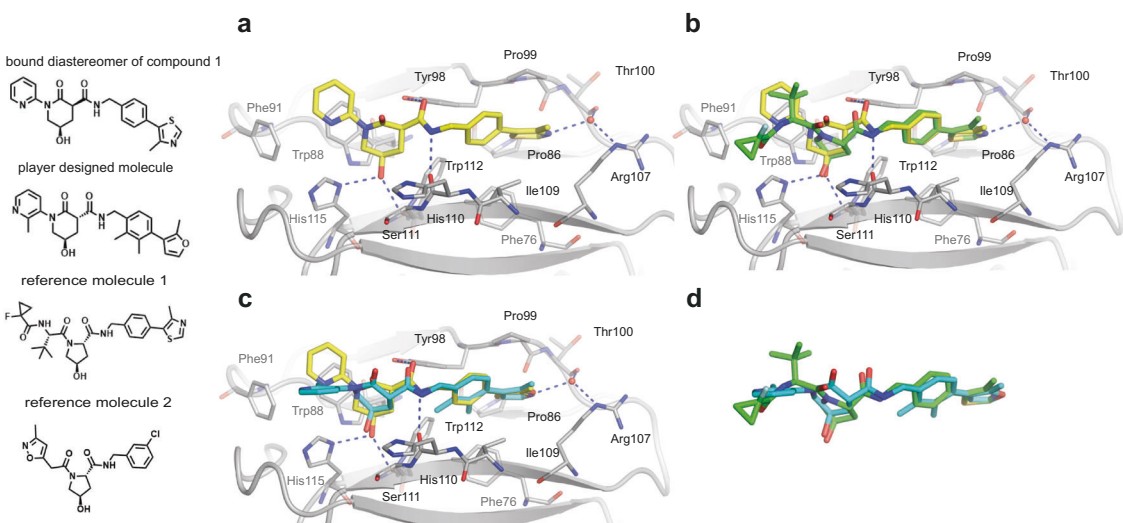

**Fig. 4 | Structural analysis. a** Binding mode of **1** as observed in the co-crystal structure with VHL (PDB code: 8P0F). **1** is color-coded by atom type with yellow carbons. **b** Superposition of the co-crystal structures of **1** (PDB code: 8P0F) and the game starting point reference molecule 1 in complex with VHL (PDB code: 5NVX, protein atoms not shown). Small molecules are color-coded by atom type with yellow and green carbons, respectively. **c** Superposition of the co-crystal structure of **1** and the original player-designed compound with the VHL protein coordinates used in the game (derived from 5NVX). Small molecules are color-coded by atom type and shown with yellow and blue carbons, respectively. **d** Superposition of the co-crystal structures of reference molecule 1 in complex with VHL and the player-designed compound. Small molecules are color-coded by atom type with green and blue carbons, respectively, and protein atoms are omitted.

## Discussion

Here, we demonstrate the feasibility of crowd-sourcing in the early stages of small-molecule drug discovery. Given an appropriate starting structure and objectives, game players—including those with limited medicinal chemistry experience—can design compounds which not only successfully meet compound quality criteria, but also bind well to a protein target of interest. By providing small molecule design tools to anyone who can download and install a game onto their computer, we expand the range of people who can contribute to drug design. Democratizing the process in this way can bring extra resources to discovering drugs for rare and neglected diseases.

Due to the provided objective criteria as well as the post-game property filtering, **1** is characterized by its improved predicted physicochemical properties, specifically with respect to the lower TPSA (Table 1). Permeability and Caco2 efflux ratio were determined at Boehringer Ingelheim. It features a higher intrinsic permeability, and lower Caco2 efflux ratio compared to reference molecule 1 (Table 1). The identified hydroxy-piperidinone core as a bio-isosteric replacement for the classical hydroxyproline motif provides an interesting starting point for further structural extension. However, the undecorated 1,3-dicarbonyl motif presents a configurationally labile stereocenter in its current state. The biophysical affinity of the diastereomeric mixture **1** ($K_D = 182\,\mu M$) toward the protein versus the peptide-like starting molecule is significantly lower; however, it should be noted that the starting molecule is the endpoint of an extensive optimization protocol. The identified, player-proposed compound (**1**) might better be viewed as a starting point for structure-based ligand optimization, therefore reference molecule 2, an early starting point for VHL ligands, was selected as benchmark (Fig. 4 and Table 1)[16]. Similar binding affinities for **1** and reference molecule 2 were observed (Table 1), particularly as the diastereomeric mixture likely leads to an underestimated affinity of the stereoisomer observed in the co-crystal structure. In addition, both molecules exhibit comparable basic molecular descriptors (Table 1), notably **1** has fewer rotatable bonds compared to reference molecule 2. A reduced number of rotatable bonds has recently been identified as an important optimization parameter for the generation of oral bioavailable PROTACs[34]. In summary, **1** is an attractive starting point for the discovery of higher potency VHL ligands with improved pharmacokinetic properties for PROTAC discovery.

The current design protocol included a fair amount of manual post-processing and evaluation. Torsional favorability played a large role in the selection process, in part due to a lack of explicit consideration during the design. The Drugit client has since been augmented to include scoring for disfavored internal rotations. Less amenable to client incorporation is the hands-on refinement of the player designs by experienced medicinal chemists. These primarily represent the removal of structural alerts or unnecessary functional groups or their replacement with the parent compound functional groups. Further investigation of tools and objectives to encourage players to perform such removal/replacements themselves is needed. Additionally, Rosetta scoring, which is used internally to evaluate the quality of binding, is not strongly correlated with binding success. Further improvement of binding evaluation in-client, potentially by incorporating other physics and machine learning-based scoring, should increase the rate of player success, as would investigating orthogonal approaches for post-game compound evaluation.

## Methods

### Puzzle setup

Ten puzzles were released in the Foldit client over consecutive weeks from Oct 20, 2021 to Jan 12, 2022. The puzzles started with the structure of the VHL domain in complex with either reference molecule 1 or a portion thereof, and varied in settings (Supplementary Tables 1, 2

and [https://doi.org/10.5281/zenodo.14902201]). During gameplay, the chemical identity and the conformation of the ligand in complex with the protein is uploaded to the server, either when a personal high score is reached, when the player chooses to upload the solution, or at regular intervals. After the conclusion of the puzzle, all ligands known to the Foldit server (high-scoring and intermediate) were collected and deduplicated, saving the best-scoring conformation of each distinct chemical identity.

### Filtering

Compounds with any atom more than 10 Å away from any starting molecule binding site atom were removed. A Pipeline Pilot 2021 (Biovia, Dassault Systèmes) workflow was devised. Player suggestions containing less than two sulfurs, less than six but at least one nitrogen, less than 30 carbons, less than five fluorines, less than five chlorines, less than two bromines, and less than two iodines were kept at first. Further restrictions included a molecular weight between more than 200 and less than 600 Da, more than 12 but below 50 atoms, less than 10 rotatable bonds, less than five rings, an ALogP below 6, a cLogP between 0.6 and 5, at least one but less than seven H-bond acceptors, and at least one but less than five H-bond donors. Compounds were flagged with a reactive SMARTS filter and filtered for quantitative estimation of drug-likeness (QED[35]) above 0.3, and a total polar surface area below 145 Å². Custom SMARTS filters were applied to remove fused benzenes, sequential alkenes, more than one alkyne, an alkyne, and a nitrile in the same molecule, aliphatic ethers in long chains, more than three amides, aliphatic chains with at least five carbons, less than three methyl groups on heteroaromatic rings, and other unwanted features. As a test for the stringency of the filters applied in Pipeline Pilot, the dataset was spiked with the 20 + VHL binders with known activity data, and the ability of these compounds to pass the filters was verified. Polybasic compounds were removed manually.

### Efflux and permeability prediction

Each efflux or permeability prediction had also an associated uncertainty. The predictions were classified and color-coded according to a traffic light scheme. For MDCK and Caco2 efflux, respectively, class A with values below 3 was "good" (green), class B with values between 3 and 10 was "moderate" (yellow), and class C above 10 was "bad" (red). For Caco2 AB permeability, class A with values above $5.0 \times 10^{-6}$ cm/s was "good" (green), class B with values in the range between $0.5–5.0 \times 10^{-6}$ cm/s was "moderate" (yellow), and class C with values below $0.5 \times 10^{-6}$ cm/s was "bad" (red).

### Docking and torsion checks

Docking was performed using Glide[36] (Glide version 2021-4, build 135, Schrödinger, Inc.). One pose per molecule was generated, minimized with an in-house tethered minimization routine, and rescored with the Glide score. A check for reasonable torsions was performed using an early version of the SMARTScompare[37] torsion quality tool, with the assumption that high-scored poses featuring "good" torsions are likely to resemble the bioactive conformation.

### Absolute binding free energy calculations

Default settings for FEP+[29,30] (Schrödinger 2021-4, Schrödinger, Inc.; https://www.schrodinger.com/products/fep) as well as the ABFE approach by Biggin and collaborators[31] were used, except for the following settings: Restraints for FEP+ were generated during a 1 ns long molecular dynamics simulation, followed by 5 ns long FEP+ runs. The OPLS4 force field[38] was used for these calculations. For the Biggin ABFE, ligands were parametrized with Parsley, the small molecule force field of the Open Force Field initiative[39]. The binding free energy was obtained via a thermodynamic cycle, in which each VHL-ligand complex leg was calculated with 44 lambda windows and each ligand-solvent leg with 33 lambda windows of 5 ns each.

## Protein expression and purification

The ternary complex of VHL (UNIPROT entry: P40337, residues 54–213, N-terminal His-Tag), EloB (UNIPROT entry: Q15370, residues 1–104), and EloC (UNIPROT entry: Q15369, residues 17–112) proteins (VCB), were co-expressed from two separate plasmids in *E. coli* strain BL21(DE3). Cells were grown in LB medium until an OD600 of 1, induced with 0.3 mM IPTG, and grown overnight at 23 °C. The cell pellet was resuspended in lysis buffer (50 mM HEPES pH 8.0, 500 mM NaCl, 0.5 mM TCEP, 1 mM MgCl$_2$, 20 mM imidazole, benzonase, and complete protease inhibitor tablets), and cells were lysed by sonication. After centrifugation and filtering, the supernatant was loaded onto a 10 mL HisTrap column (Cytiva), washed with 50 mM HEPES pH 8.0, 500 mM NaCl, 0.5 mM TCEP, 20 mM imidazole until the UV baseline was stable and protein was eluted with the same buffer containing 500 mM imidazole. The elution fraction was dialyzed against 20 mM HEPES pH 8.0, 20 mM NaCl, 1 mM DTT, and loaded onto a Resource Q column (Cytiva) equilibrated in the same buffer. Protein was eluted with a linear salt gradient from 20 to 500 mM NaCl over 40 column volumes. Fractions containing the VCB complex were pooled, concentrated, and loaded onto a Superdex 200 26/60 size exclusion column (Cytiva) equilibrated with 20 mM HEPES pH 7.5, 100 mM NaCl, and 1 mM TCEP. The final purified complex was concentrated at 30 mg/mL, flash-frozen, aliquoted, and stored at −70 °C. The protein yield was 10 mg/L of expression. For the expression of selective [$^{13}$C-methyl]-Val-Leu-Met-labeled VHL, M9 minimal medium supplemented with [$^{13}$C-methyl] alpha-keto-isoValerate (0.1 g/L) and [$^{13}$C-methyl] methionine (0.1 g/L) was prepared. The medium was inoculated with transformed *E. coli*, induced with IPTG (0.25 mM) at an optical density (OD600) of 0.5, and grown overnight at 23 °C. The protein was purified as described above for the unlabeled protein.

## Compound synthesis

Full synthesis procedures of compounds used in this study is described in Supplementary Methods.

## TR-FRET VCB−VHL tracer binding assay

A time-resolved fluorescence resonance energy transfer (TR-FRET) VCB−VHL Tracer binding assay assay was used to identify compounds which inhibit the binding of a Cy5-labeled VHL binder (tracer) to VCB. His-tagged VCB complex corresponding to VHL amino acids 54–213 with N-terminal His-tag and TEV cleavage site in complex with EloB amino acids 1–104 and EloC amino acids 17–112 were co-expressed in *E. coli*. Cy5-labelled VHL Tracer analog to ref. 32 using different linker lengths and Cy5 as TR-FRET acceptor was used as VCB binding partner in the assay. Test compounds dissolved in DMSO were dispensed onto assay plates (Proxiplate 384 PLUS, white, PerkinElmer; 6008289) using an Access Labcyte Workstation with the Labcyte Echo 55x. For the chosen highest assay compound concentration of 100 μM, 150 nL of compound solution was transferred from a 10 mM DMSO compound stock solution. A series of eleven fivefold dilutions per compound was transferred to the assay plate. Compound dilutions were tested in duplicates. DMSO was added as a backfill to a total volume of 150 nL. The assay runs on a fully automated robotic system. 15 nL of the Cy5-labeled VHL Tracer (10 μM stock in 100% DMSO) was added to rows 1–23 using the Labcyte Echo 55x for transfer. About 15 nL of 100% DMSO was added to row 24. 15 μL of VCB reaction mix in assay buffer (20 mM HEPES pH 7.3, 150 mM NaCl, and 0.005% Tween 20) including His-tagged VCB complex (10 nM final assay concentration) and Eu-anti 6xHis antibody (PerkinElmer AD0110, 3 nM final assay concentration) was added to rows 1-24. Plates are kept at room temperature. After 40 min incubation time, the signal is measured in a PerkinElmer Envision HTS Multilabel Reader using the TR-FRET LANCE Ultra specs from PerkinElmer. Each plate contains 16 wells of a negative control (diluted DMSO instead of a test compound; column 23 with Cy5-labeled VHL Tracer) and 16 wells of a positive control (diluted DMSO instead of a

test compound; column 24 without Cy5-labeled VHL Tracer). As an internal control, an unlabeled VHL Tracer can be measured on each compound plate. IC$_{50}$ values are calculated and analyzed with Boehringer Ingelheim's MEGALAB IC$_{50}$ application using a 4-parametric logistic model.

## $^{19}$F reporter assay and NMR detected solubility

The reporter assay was set up as described previously, with compound 19 from the paper as reporter[33]. In short, 2 μM VCB (VHL-ElonginC-ElonginB) complex, 50 μM reporter, and 500 μM of the respective compound to be tested were used in all competition experiments. The cpmg delay was set to 40 ms, and two T2 relaxation delays were recorded with relaxation times of 80 and 480 ms, respectively. The experiment with the 80 ms delay was used to report the % recovery values shown in Table S6. The % recovery was calculated by using the integral of the $^{19}$F signal of the reporter in the absence of competitor as 0% recovery, the integral of the reporter in the absence of protein and competitor as 100% recovery, and the integral of the reporter in presence of both protein and competitor compounds as % recovery. In addition, we performed a solubility check of the compounds to be tested in the $^{19}$F competition experiment. Here, we set out to make a 500 μM solution of the compound in the NMR buffer consisting of 20 mM TRIS and 100 mM NaCl at pH 7.5 in D$_2$O (not corrected). Integration of the compounds $^1$H NMR signals was compared to 250 μM maleic acid as an internal standard, which then yielded the solubility values reported in Table S6 as only the soluble part of the ligands appears in the NMR spectrum. Compounds showing no signal intensity at all were categorized with a solubility of <10 μM. For some of the compounds, we then refrained from measuring the competition with the reporter as they were completely insoluble in the NMR buffer tested, which would potentially lead to artifacts.

## Solubility testing

Compound solubility was determined by dilution of a 10 mmol/L compound solution in DMSO into buffer to a final concentration of 125 μg/mL. Dilution into a 1:1 mixture of acetonitrile and water was used as a reference. After 24 h, the incubations were filtrated, and the filtrate was analyzed by LCUV.

## Bidirectional permeability in Caco2 Cells

Bidirectional permeability of test compounds across a Caco2 cell monolayer was measured as described previously[40]. Briefly, compounds were diluted in transport buffer (128.13 mM NaCl, 5.36 mM KCl, 1 mM MgSO$_4$, 1.8 mM CaCl$_2$, 4.17 mM NaHCO$_3$, 1.19 mM Na$_2$HPO$_4$, 0.41 mM NaH$_2$PO$_4$, 15 mM 2-[4-(2-hydroxyethyl)piperazin-1-yl]ethanesulfonic acid (HEPES), 20 mM glucose, 0.25% bovine serum albumin, pH 7.4) to a final concentration of 1 or 10 μM and added to the apical or basolateral (donor) compartment. Cells were incubated with the compounds for up to 2 h, and samples from the opposite (receiver) compartment were taken at different time points.

## Protein-based NMR hit validation

Protein NMR experiments were performed on a Bruker 700 MHz spectrometer equipped with a 5 mm TCI cryoprobe in 3 mm NMR tubes filled with 180 μl of sample. 20 mM TRIS buffer at pH 7.5 and a 100 mM NaCl salt concentration, supplemented with 10% D$_2$O for locking, was used for all experiments. Compound concentration was 500 μM for single point measurements and specifically $^{13}$C-methyl labeled (Met, Ile, Leu, Val) VCB complex was measured at concentrations of 100 and 200 μM to validate binding and determine the $K_D$[41].

## Protein crystallization and structure solution

Compound (**1**) and the VCB complex (10 mg/mL in 20 mM HEPES pH 7.5, 100 mM NaCl, 1 mM TCEP) were incubated for 1 h at 4 °C. For crystallization, 250 nL of the complex were mixed with 160 nL

reservoir solution (20% PEG 4000, 200 mM MgCl$_2$ and 100 mM TRIS pH = 7.5) in 96-well sitting drop plates. Brick-shaped crystals of ~100 μM length grew after 4 days at room temperature. Crystals were flash-frozen in liquid nitrogen using the reservoir buffer with 25% ethylene glycol as a cryo-protectant. Data were collected at the SLS beamline PXII and processed using autoPROC[42] and STARANISO[43]. Resolution cut-offs were determined using the default setting in STARANISO. The structure was solved by molecular replacement using the PDB structure 1VCB as a search model. The model was refined in alternating cycles of Coot[44] and autoBUSTER (Global Phasing Ltd., Cambridge, United Kingdom, 2017).

### Reporting summary

Further information on research design is available in the Nature Portfolio Reporting Summary linked to this article.

## Data availability

All player-design compounds, with coordinates corresponding to their highest-scoring poses have been deposited in the Zenodo database: https://doi.org/10.5281/zenodo.14902201. The X-ray crystallography data for **1** in complex with VHL generated in this study has been deposited in the wwPDB database under accension code 8P0F The previously published starting structure of VHL is available in the wwPDB under accension code 5NVX. Source data are provided with this paper.

## Code availability

The Drugit/Foldit client can be downloaded from https://drugit.org or https://fold.it. Because Foldit crowdsourcing relies on regulated, fair competition between participants, the source code of the Foldit user interface is not open. The underlying Rosetta macromolecular modeling suite (https://www.rosettacommons.org) and Foldit Standalone, the research version of Foldit is freely available to academic and non-commercial users, and commercial licenses are available via the University of Washington CoMotion Express License Program.

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

## Acknowledgements

We dedicate this publication to the memory of Christian Alan Paul Smethurst. We thank Alessio Ciulli for consultancy and providing the [19]F reporter molecule. Work in the Meiler laboratory is supported through the NIH S10 OD016216 (J.M.), NIH S10 OD020154 (J.M.), and NIH S10 OD032234 (J.M.). J.M. acknowledges funding by the Deutsche Forschungsgemeinschaft (DFG) through SFB1423 (421152132), SFB 1052 (209933838), and SPP 2363 (460865652). J.M. is supported by a Humboldt Professorship of the Alexander von Humboldt Foundation. J.M. is supported by BMBF (Federal Ministry of Education and Research) through the Center for Scalable Data Analytics and Artificial Intelligence (ScaDS.AI). This work is partly supported by BMBF (Federal Ministry of Education and Research) through DAAD project 57616814 (J.M. SECAI, School of Embedded Composite AI) and by a grant from Boehringer Ingelheim (R.M.). C.A.P.S., Y.W., M.M., P.G., R.K., T.B., G.B., Z.J., P.S.S., J.E.F., A.M., C.H., T.G., A.B., H.W., D.B.McC., and J.B. were full-time employees of Boehringer Ingelheim at the time this study was performed. Participation in the Foldit project is overseen by the University of Washington, under STUDY00001238 "Scientific Discovery Games". Participants must agree to the IRB-certified "Foldit Terms of Service and Consent" prior to playing.

## Author contributions

T.S., S.A.C., R.P., S.P., S.K., and R.M. developed the Drugit software. T.S., C.A.P.S., and R.M. developed puzzle settings, oversaw gameplay, and collected results. F.P. designed compounds. C.A.P.S., P.G., T. B., and C.H. examined suggested molecule and designed synthetic strategies Y.W., Z.J., A.B., P.S.S., and A.M. performed post-competition filtering and absolute binding free energy calculations M.M. and RK. performed protein NMR and small molecule analytics, G.B. performed x-ray structure analysis, T.G. performed TR-FRET assays, J.E.F., A.G.W., and H.W., provided guidance during the project. C.A.P.S, J.B., and R.M. lead the respective teams of the collaboration and D.B.McC. and J.M. provided concept, supervision, and funding. Y.W., T.B., T.S., J.B., and R.M. prepared the manuscript with input from all authors.

## Competing interests

The authors declare no competing interests.

## Additional information

**Thomas Scott**[1,2,12], **Christian Alan Paul Smethurst**[3,12,13], **Yvonne Westermaier** [3], **Moriz Mayer**[3], **Peter Greb** [3], **Roland Kousek**[3], **Tobias Biberger**[3], **Gerd Bader** [3], **Zuzana Jandova**[3], **Philipp S. Schmalhorst** [3], **Julian E. Fuchs** [3],

Aniket Magarkar [4], Christoph Hoenke[4], Thomas Gerstberger [3], Steven A. Combs[1,2], Richard Pape[1,2], Saksham Phul [1,2], Sandeepkumar Kothiwale[1,2], Andreas Bergner[3], Foldit Players*, Alex G. Waterson[1,5,6], Harald Weinstabl [3], Darryl B. McConnell [3], Jens Meiler [1,2,5,7,8,9,10,11] ✉, Jark Böttcher [3] ✉ & Rocco Moretti [1,2] ✉

[1]Department of Chemistry, Vanderbilt University, Nashville, TN 37235, USA. [2]Center for Structural Biology, Vanderbilt University, Nashville, TN 37240, USA. [3]Boehringer Ingelheim RCV, Dr. Boehringer-Gasse 5-11, 1121 Vienna, Austria. [4]Boehringer Ingelheim Pharma GmbH & Co. KG, Birkendorfer Str. 65, 88397 Biberach an der Riß, Germany. [5]Departments of Pharmacology, Vanderbilt University, Nashville, TN 37235, USA. [6]Vanderbilt Institute of Chemical Biology, Nashville, TN 37232, USA. [7]Center for Applied Artificial Intelligence in Protein Dynamics, Vanderbilt University, Nashville, TN 37240, USA. [8]Institute of Drug Discovery, Faculty of Medicine, University of Leipzig, 04103 Leipzig, Germany. [9]Faculty of Mathematics and Informatics, University of Leipzig, 04103 Leipzig, Germany. [10]Faculty of Chemistry and Mineralogy, University of Leipzig, 04103 Leipzig, Germany. [11]Germany Center for Scalable Data Analytics and Artificial Intelligence and School of Embedded Composite Artificial Intelligence SECAI, Dresden/Leipzig, Germany. [12]These authors contributed equally: Thomas Scott, Christian Alan Paul Smethurst. [13]Deceased: Christian Alan Paul Smethurst. *A list of authors and their affiliations appears at the end of the paper. ✉e-mail: jens@meilerlab.org; jark.boettcher@boehringer-ingelheim.com; rocco.moretti@vanderbilt.edu

## Foldit Players

Aleshin Dmitry Alekseevich[14], Walter Barmettler[14], Krzysztof Bogusz[14], Alexander Boykov[14], Andrew Cai[14], Jeffrey Michael Canfield[14], Andy M. Chen[14], Keith Clayton[14], Alan Roger Coral[14], Martin Dickel[14], Harald Feldmann[14], Lezhe Gao[14], Thomas J. George[14], Kevin M. Gildea[14], Gregory T. Hansen[14], Alyssa Joy Higgins[14], Renton Braden Mathew Innes[14], Bruno Kestemont[14], Allen Lubow[14], George Victor McIlvaine[14], Ian James McNaughton[14], Cédric Partinico[14], Vincent Pit[14], Alexa Sen[14], Michael Simon[14], Vera Simon[14], Robert Spearing[14], Bora Tastan[14], Stephen Vincent[14], Linda Wei[14], Jonathan Weinberg[14], Kevin D. Wells[14], Douglas Craig Wheeler[14], Ulas Yeginer[14], Scott J. Zaccanelli[14] & Carlos Zambrano-Bukonja[14]

[14]Unaffiliated: Aleshin Dmitry Alekseevich, Walter Barmettler, Krzysztof Bogusz, Alexander Boykov, Andrew Cai, Jeffrey Michael Canfield, Andy M. Chen, Keith Clayton, Alan Roger Coral, Martin Dickel, Harald Feldmann, Lezhe Gao, Thomas J. George, Kevin M. Gildea, Gregory T. Hansen, Alyssa Joy Higgins, Renton Braden Mathew Innes, Bruno Kestemont, Allen Lubow, George Victor McIlvaine, Ian James McNaughton, Cédric Partinico, Vincent Pit, Alexa Sen, Michael Simon, Vera Simon, Robert Spearing, Bora Tastan, Stephen Vincent, Linda Wei, Jonathan Weinberg, Kevin D. Wells, Douglas Craig Wheeler, Ulas Yeginer, Scott J. Zaccanelli, Carlos Zambrano-Bukonja.

