## [Transparent Peer Review file · Nature Communications]

Drugit: Crowd-sourcing molecular design of non-peptidic VHL binders

Corresponding Author: Dr Rocco Moretti

Version 0:

Reviewer comments:

Reviewer #1

(Remarks to the Author)

This article by Scott, Smethurst et al. discusses the introduction of Drugit, a small molecule design tool of the online citizen science game Foldit, as concept to involve non-expert citizens into small molecule design in early drug-discovery projects. As test case a Drugit puzzle series aimed at designing novel VHL E3 ligase ligands with improved pharmacokinetic properties, e.g. by replacing the hydroxyproline (Hyp) core motif of common VHL ligands, is presented. After completion of all puzzle rounds, compounds submitted by the citizen players have been filtered and computationally and manually evaluated to identify promising candidates, which were synthesised and tested. Compound 1 was biophysically identified as novel, non-Hyp based binder of VHL and its binding mode to VHL was structurally evaluated.

While the Drugit platform presents an exciting opportunity to involve non-scientists in early drug discovery stages and the VHL puzzle series is a suitable test case to explore the potential of compounds designed by non-experts using this gamified approach, the quality of the data currently presented in the manuscript and the supporting information does not meet the publisher's criteria for publication. In particular, experimental data is incomplete and experimental methods have not been described in either manuscript or supporting information, rendering a proper evaluation of experimental methods and results impossible.

Detailed discussion:

Introduction:

- While VHL's important role for the TPD field and the design of PROTACs has been highlighted, its biological function/potential effects of inhibition/potential therapeutic use have not been mentioned at all. As this publication focusses on novel binders for VHL, a brief mention of its biology might be worthwhile.
- Line 59: "the original molecule": Which molecule is this referring to? Extensive design efforts have been made after 2012 (cited in reference 18), improving IC50 from ~5 μ M to ~40 nM (e.g. VH101) for VHL binders. The wording "small variations" does not reflect the considerable optimisation efforts made after the initial report of a small molecule VHL binder.
- Lines 60 to 62: Reference 20 seems unsuitable to support the argument: This publication discusses stereoelectronic effects and specific binding of C4-exo/endo prolyl conformations of Hyp to VHL, but neither VHL-binder optimisation ("de-peptidising") nor physicochemical properties of the Hyp core.
- Line 61: the core motif of established VHL ligands is a hydroxyproline, not hydroxyprolinol.

Game design:

- Apart for summarising the objectives of each round of the game in table S1, the manuscript would benefit greatly from presenting a (med-chem) rationale for the choice of these objectives, including set limits.
- Could the authors please discuss if there has been a reasoning for keeping the protein atoms fixed in the puzzle? As evidenced by co-crystal structures of different binders with VHL (e.g. PDB: 4W9H vs 5NVX), some side chains adopt different conformations to accommodate different bound small molecules. Fixing the protein atoms in the position of the VH101-bound co-crystal structure (PDB: 5NVX) might affect binding scores of molecules not matching the shape of VH101.

Post-Competition Filtering:

- Line 184: "including known VHL binders 2 and 10" – while compound 10 was introduced earlier, "VHL binder 2" has not been introduced, there is no reference associated with this compound, nor a structure given within the manuscript. Further

information/references to this compound have to be included.

- Lines 184 – 186: Please add computed binding affinities of the final computational assessment (for example as table in SI).
- A majority of the prioritized molecules for synthesis feature a 2-methyl-4-phenylthiazole motif (which is common to all potent VHL ligands). Does this originate from the player-designed molecules, or was it reintroduced in the post-game filtering and adapting steps? It would be highly informative to add a table comparing the initial game-output molecules of this final selection with the slightly optimised/modified ones which were synthesised and tested. So far, the extent of mentioned “minor adaptation” of the in-game designed molecules prior to synthesis is unknown.

In vitro analysis performed on compound 1:

- 19F displacement assay: Have these experiments been performed with diastereomeric mixtures or enantiopure compound? Please discuss along the results, as it might imply higher potency of the eutomer.
- 19F displacement assay: While “different degrees of displacement” have been observed in the initial compound screen at 50 μ M, only compound 1 showed dose-dependent displacement. Were these dose-dependent experiments performed using the same assay conditions? Could the authors comment on why only compound 1 showed dose-dependent displacement, but any of the others? Is this observation in line with the initial screen at 50 μ M (viz. was compound 1 the only compound showing significant displacement in the initial screen as well)?
- An experimental direct comparison of compound 1 to VH101 and a first-generation VHL binder (see: JACS 2012, 134, 4465-4468; Angew. Chem. Int. Ed. 2012, 51, 11463 –11467) via the protein-observed NMR titration protocol would be highly recommended.
- NMR experiments: As dose-dependent experiments for compound 1 have been performed via ligand observed displacement assay and protein-observed titrations, the K_i (inhibition constant) value from the displacement assay could be used to validate the K_D value derived from the protein-observed experiments.
- K_D determination: Results should stem at least from independent duplicates of the performed assays.
- The data of the mentioned TR-FRET assay (see lines 207-210) is missing, both in manuscript and SI. Furthermore, the procedure of this assay is missing as well.
- The determined IC_{50}/K_D values of the novel compound 1 should be compared to those of established VHL binders. Taking into account that compound 1 represents an un-optimised first hit, it would be recommended to compare these values not only to VH101 (“VHL binder 10”), but also to early hits from the structure-guided optimisation process (see e.g. JACS 2012, 134, 4465-4468; Angew. Chem. Int. Ed. 2012, 51, 11463 –11467). This comparison should also be referred to in the later discussion on the usability of Drugit to identify molecular starting points for ligand optimisation.
- Table 1 shows in vitro ADME properties. These have not been mentioned at any point in the manuscript, nor in the SI and no protocol/procedure is given for these assays. If these assays have been performed, they should be included in the discussion of the in vitro analysis of compound 1 and details have to be added to the SI.

Co-crystal structure:

- How does the protein structure of the co-crystal of compound 1 with VHL compare with the VHL protein coordinates used in the game – are there significant changes in sidechain conformations? This comparison could be very informative on the impact of keeping the protein atoms fixed in the Drugit framework, and how this might affect the hit-finding process.
- As the result of the co-crystal structure with compound 1 in (R,R)- configuration does not match the configuration of the original player-designed ((R,S)- configuration) it would be very interesting to calculate binding affinity (see lines 183-186) of both conformations for the original player-designed compound and the post-game optimised compound 1 in direct comparison.

Discussion:

- The here presented game designs novel ligands for a protein target based on co-crystal structure of a highly potent ligand of a known binding site, resulting in a >600-fold less potent novel binder (comparing K_D values). It would be appreciated if the authors could comment on the productivity/efficiency of this gamified approach of finding novel binders (would experts design better hit candidates from scratch?) and to identify binders for less-explored targets.
- Lines 250-253: The novelty of compound 1 is arguably exchanging Hyp with the hydroxypiperidinone core motif. If the idea is to replace this hydroxypiperidinone core motif in later structure-based ligand optimisation, wouldn't structure-based ligand optimisation from a more potent Hyp-containing binder (with the goal to replace the Hyp motif) be a more efficient approach?

Referral between figures and text:

In several cases, there is a disconnect between the description of figure subsections in the main text and the content of these subsections, such as

- Figure 1a depicts the atom selection tool of the Small-Molecule Design panel of the Drugit interface, but in the main text Figure 1a is described as the main window showing the “3D structure of a starting molecule docked” (lines 76-78). This view is in fact presented in Figure 1c.
- Figure 1d shows the undo function of Drugit but is referred to as Figure 1c in the main text (line 110).
- The description of figure 2b in the main text (lines 156-158) is misleading, as all 6,500 compounds docked to VHL are shown here, not the reduced library of 19 compounds.
- There is no mention of Figure 2c in the main text. A discussion of the chemical space of in-game designed compounds vs current VHL inhibitors would be highly desirable. In the current version, Figure 2c does not seem to have any purpose. How has the PCA been performed? There is no description of the PCA in either the manuscript or the SI.

Figures & Tables:

- Table 1 lacks a description. References have to be included for experimental values of “small molecule 10” taken from earlier reports.
- Figure 3a: The colours in the 1D spectrum of (a) are not clearly identifiable. I would recommend increasing the line width of

the signals for better distinguishability.

- Figures 3: In the description of (a), the colour for the signal after addition of compound 1 should be included.
- Figure 4(a,b): As the discussion refers to potential optimisation of compound 1 by engaging with Trp88 and Phe91 via π - π interactions, it would be highly recommended to select a view of the binding site of VHL that includes both residues. Currently, Phe91 is not shown.
- Figure 4b: is the protein structure derived from PDB: 8P0F (VHL w/ 1) or 5NVX (VHL w/ "compound 10" (VH101))?

Supporting Information:

- Table S1: "no center: small molecule 10 lacking the hydroxyproline sidechain": Does that mean replacement of Hyp with glycine? If so, please state more clearly.
- Please add protocols/procedures for the ^{19}F displacement screen as well as dose-dependent assay, TR-FRET assay, determination of aqueous solubility and Caco-2 permeability, expression/labelling of ^{13}C methyl labelled (Ile, Leu, Val) VCB complex.
- Include results of TR-FRET assay.
- Add information (table) on %displacement of all tested compounds of the initial NMR displacement assay.
- The section on general synthetic methods, materials (e.g. sources for commercial compounds) and instrumentation is missing. Please check author guidelines on the methods section on required information.
- References for already known compounds are missing.
- Yields in mmol, Rf values and melting points missing in the compound characterisation data.
- LC-MS results: Calculated m/z of the $[\text{M}+\text{H}]^+$ has to be added in the compound characterisation data. Is the accuracy of the used instrument really $m/z \pm 1$, viz. are all significant decimals shown?
- Characterisation of chiral compounds is completely missing. This includes: Details on chiral SFC purification in the synthesis section (e.g. for compounds 20 and 21), determination of $[\alpha]_D$ values, characterisation using appropriate techniques, including polarimetry, NMR, single crystal XRD, or by correlation of HPLC or GC (see characterisation of chiral compounds section in author guidelines).
- Information on purity of compounds is missing. Purity cannot be checked as NMR spectra aren't attached.

Further small corrections:

- Line 31: "designed by a player": This wording in the abstract seems a bit misleading – while the core hydroxypiperidinone structure has been designed by a player, its eastern and western extension have been substantially modified in post-game processing.
- I would recommend referring to "small molecule 10" as VH101, which is the more known (and less ambiguous) term for this molecule. Regardless of this, the naming of this compound should be consistent throughout the manuscript – currently it is referred to as "small molecule 10", "VHL binder 10" and "ligand 10". It might be beneficial to add the chemical structure of VH101 to Figure 2a.
- I would recommend to either use "western" and "eastern" or "left-hand side" and "right-hand side" to refer to different parts of the small molecule, instead of mixing these terms.
- I would recommend using consistent nomenclature to refer to "larger than" – either "260+" or ">260".

Reviewer #2

(Remarks to the Author)

The authors describe development of "Drugit", a small molecule design mode of popular citizens science game Foldit. The authors then test the ability of users to design a binder to VHL, a popular ligase in the TPD field and after extensive filtering of designs arrive at a 250 μM binder designed by a non-medical chemist user.

This is a clearly written and concise story on how citizens science can help in guiding the development of new molecules for challenging and yet interesting targets such as VHL.

I would recommend to publish this report in its current form. My only suggestion is to add a link to the website where users can play Drugit.

Reviewer #3

(Remarks to the Author)

This is a quite an original paper investigating whether "citizen scientists", by using an on-line simple tool ("Foldit"), manage to identify drug leads. One of the molecules identified by a pool of people with no background in medicinal chemistry turned out to bind to the chosen protein target (a ligase); the subsequent X-ray structure showed that a binding mode similar to that predicted by the citizen scientist. The approach suggests that citizen scientists could be involved in the highly costly process of structure-based drug design. This work is of high significance in the field of computer-aided drug design. It uses a sound methodology and it can be easily reproduced (especially the on-line applications). The paper is nice to read and very well organized.

Minor points:

- The societal impact of the work could be added.
- The limitations of the on-line docking method relative to the state-of-the-art could be summarized.
- The knowledge required to run the on-line applications should be stated.
- Enzymes are relatively easy targets: could this approach be used, for instance, to allosteric ligands of GPCRs or ion

channels?

Reviewer #4

(Remarks to the Author)

One of the most important problems in the application of molecular biophysics to medicine is drug design. The authors address early stages of the process by crowdsourcing through a game, Drugit, inspired by and building on Foldit, a broadly known application of Rosetta energy function to protein folding and design. This particular study presents a single test case based on VHL E3 ligase: from posing the task to experimental validation of the designed molecule by X-ray crystallography, among other methods, thus demonstrating the success of the design. Although promising, it is important to remember that this is an early (and possibly the simplest) stage of drug design, and 90% of the road is still ahead (which may be good to mention in the manuscript).

A more general question is whether Rosetta energy function is a good indicator of success in drug design. The authors indicate that it might not be. What are possible alternatives?

Using “objectives” is a nice way to address the usual problems in drug design that are not related to binding energy.

Overall, this work presents a detailed description of the experiment and should be interesting to the readers. Although I remain a bit skeptical whether the Drugit module holds a promising future in drug design, the experiment presented in the paper is well-designed and well-presented. It is difficult to suggest any improvements to this work as it stands, without significant reconsideration.

Just out of curiosity, the protein selected for the experiment is rather small, and probably smaller than a typical drug target. Any issues with the Drugit game/players to work with larger targets?

Small matters:

1. Consistency in punctuation: ‘Drugit’ (quotes of different style).
2. Maybe it is better to use “dimension 1” and “2” instead of “0” and “1” in Fig. 2? These are known as the first and the second principal components. There is no 0th component. You already use “compound 1” and not “compound 0”.
3. Fig. 2 might be better to use “player-designed” with a hyphen.

Version 1:

Reviewer comments:

Reviewer #4

(Remarks to the Author)

The authors carefully addressed my comments and I have no further questions about this work. I also studied the comments by other reviewers and revisions done by the authors. I think the revision is commendable and it clearly improved the manuscript. The authors took all the comments seriously and performed additional, sometimes quite laborious, experiments, which support the conclusions. The work is sound, presented well, and adds to our understanding of protein modeling and drug design.

(Remarks on code availability)

REVIEWER COMMENTS

Reviewer #1 (Remarks to the Author):

This article by Scott, Smethurst et al. discusses the introduction of Drugit, a small molecule design tool of the online citizen science game Foldit, as concept to involve non-expert citizens into small molecule design in early drug-discovery projects. As test case a Drugit puzzle series aimed at designing novel VHL E3 ligase ligands with improved pharmacokinetic properties, e.g., by replacing the hydroxyproline (Hyp) core motif of common VHL ligands, is presented. After completion of all puzzle rounds, compounds submitted by the citizen players have been filtered and computationally and manually evaluated to identify promising candidates, which were synthesized and tested. Compound 1 was biophysically identified as novel, non-Hyp based binder of VHL and its binding mode to VHL was structurally evaluated.

While the Drugit platform presents an exciting opportunity to involve non-scientists in early drug discovery stages and the VHL puzzle series is a suitable test case to explore the potential of compounds designed by non-experts using this gamified approach, the quality of the data currently presented in the manuscript and the supporting information does not meet the publisher's criteria for publication. Experimental data is incomplete and experimental methods have not been described in either manuscript or supporting information, rendering a proper evaluation of experimental methods and results impossible.

We thank the reviewer for the comprehensive and detailed review which helped us to significantly improve the manuscript. In the following you will find comments, corrections, and additions to the manuscript.

Detailed discussion:

Introduction:

- While VHL's important role for the TPD field and the design of PROTACs has been highlighted, its biological function/potential effects of inhibition/potential therapeutic use have not been mentioned at all. As this publication focusses on novel binders for VHL, a brief mention of its biology might be worthwhile.

We would like to thank the reviewer for highlighting this. We added a respective section "VHL is a component of the CUL2-RBX1-ElonginB-ElonginC-VHL cullin-RING ubiquitin ligase complex, and its natural substrates are hypoxia-inducible factor (HIF) proteins hydroxylated at a conserved proline amino acid by prolyl-hydroxylase domain proteins "

- Line 59: "the original molecule": Which molecule is this referring to? Extensive design efforts have been made after 2012 (cited in reference 18), improving IC₅₀ from ~5 μM to ~40 nM (e.g. VH101) for VHL binders. The wording "small variations" does not reflect the considerable optimization efforts made after the initial report of a small molecule VHL binder.

The sentence has been rephrased to avoid unintended implications: "While extensive optimization has been performed on molecules originally introduced by Crews and Ciulli, including progress in 'de-peptidizing' parts of the molecule to reduce polarity, the common hydroxyproline core crucial for binding affinity carries unfavorable physicochemical properties. "

- Lines 60 to 62: Reference 20 seems unsuitable to support the argument: This publication discusses stereo-electronic effects and specific binding of C4-exo/endo prolyl conformations of Hyp to VHL, but neither VHL-binder optimization (“de-peptidizing”) nor physicochemical properties of the Hyp core.

We agree and remove the citation.

- Line 61: the core motif of established VHL ligands is hydroxyproline, not hydroxyprolinol.

Corrected accordingly

Game design:

- Apart from summarizing the objectives of each round of the game in table S1, the manuscript would benefit greatly from presenting a (med-chem) rationale for the choice of these objectives, including set limits.

The text has been modified as follows: “The preliminary results from each round were examined by medicinal chemists, and set limits adjusted according to in-house criteria and expert opinion on the quality of observed compounds.”

- Could the authors please discuss if there has been a reasoning for keeping the protein atoms fixed in the puzzle? As evidenced by co-crystal structures of different binders with VHL (e.g. PDB: 4W9H vs 5NVX), some side chains adopt different conformations to accommodate different bound small molecules. Fixing the protein atoms in the position of the VH101-bound co-crystal structure (PDB: 5NVX) might affect binding scores of molecules not matching the shape of VH101.

Ultimately, keeping the protein atoms fixed was a judgment call and the rationale has been added to the manuscript: “However, for this work it was decided to hold protein atoms fixed and only allow the small molecule to move. This more closely matches the prevalent molecule design approach in early-stage drug discovery and should focus players efforts on compounds which match the binding mode of the starting point. More diverse compounds might be found with future experiments using proteins with more flexibility.”

Post-Competition Filtering:

- Line 184: “including known VHL binders 2 and 10” – while compound 10 was introduced earlier, “VHL binder 2” has not been introduced, there is no reference associated with this compound, nor a structure given within the manuscript. Further information/references to this compound have to be included.

We would like to thank the reviewer for highlighting this. We removed the statement regarding VHL binder 2 as our reference point throughout the manuscript is ligand 10 (now referred to reference molecule 1 to increase clarity). We included the calculated values in Table S7.

- Lines 184 – 186: Please add computed binding affinities of the final computational assessment (for example as table in SI).

Table S4 has been added, which shows the player designed compounds and related, synthetically more accessible compounds for which ΔG values were calculated by FEP+. For VHL ligands,

absolute binding free energy predictions obtained by FEP+ were more consistent than with the ABFE method by Biggin *et al.*, which is why the latter values are not shown.

- Most of the prioritized molecules for synthesis feature a 2-methyl-4-phenylthiazole motif (which is common to all potent VHL ligands). Does this originate from the player-designed molecules, or was it reintroduced in the post-game filtering and adapting steps? It would be highly informative to add a table comparing the initial game-output molecules of this final selection with the slightly optimized/modified ones which were synthesized and tested. So far, the extent of mentioned “minor adaptation” of the in-game designed molecules prior to synthesis is unknown.

Many thanks for this excellent suggestion. We added Table S3, pairing the molecules as designed by players with the corresponding structures which have been synthesized and tested.

In vitro analysis performed on compound 1:

- ¹⁹F displacement assay: Have these experiments been performed with diastereomeric mixtures or enantiopure compound? Please discuss the results, as it might imply higher potency of the eutomer.

All experiments have been performed with the diastereomeric mixture; we added the information more clearly in the manuscript. We included the respective results in the discussion.

- ¹⁹F displacement assay: While “different degrees of displacement” have been observed in the initial compound screen at 50 μM, only compound 1 showed dose-dependent displacement. Were these dose-dependent experiments performed using the same assay conditions? Could the authors comment on why only compound 1 showed dose-dependent displacement, but any of the others? Is this observation in line with the initial screen at 50 μM (viz. was compound 1 the only compound showing significant displacement in the initial screen as well)?

We would like to thank the reviewer for highlighting this disconnect, indeed all ¹⁹F experiments were performed at a single concentration of 500 μM compound. During editing of the manuscript there was a misunderstanding of Figure 3a, which represents different molecules and not a titration. We apologize for the error and have rewritten the section based on your suggestions and those of the additional reviewers. The profiling data for all synthesized compounds was added to the supplementary section.

“Synthesized compounds were submitted to a fluorescence resonance energy transfer (TR-FRET) assay using a Cy5-labeled VHL Tracer analog³², an NMR based solubility assay and a ¹⁹F NMR displacement assay³³ (Supplementary Table 6), The later one is a highly sensitive technique to confirm that the compounds bind to the protein pocket of interest. Here, a well characterized ¹⁹F-containing reporter molecule was used at 50 μM concentration in presence of 2 μM VCB (VHL-ElonginC-ElonginB) complex. Addition of the compounds to be tested at 500 μM concentration led to different degrees of displacement of the reporter probe from the binding site, and therefore to reappearance of the ¹⁹F NMR signal in the spectrum (Figure 3a). The diastereomeric mixture of compound 1 was selected for further profiling as it showed a competitive behavior in the TR-FRET assay (IC₅₀ = 264 ± 30 μM), excellent solubility (>500 μM) and a dose-dependent displacement of the ¹⁹F NMR probe (52% recovery). Compound 1 was submitted to protein-observed NMR experiments: Protein labeled selectively, with ¹³C methyl groups in the residues Ile, Val, Leu, and Met was used to obtain a K_D = 182 ± 76 μM by fitting the dose-dependent shifts (Figure 3b and c). The shift pattern induced by compound 1 is very similar to the ones induced by published VHL ligand VH298 and an initial starting point in VHL

ligand discovery (named reference molecule 2 in this study) (Table 1 and Figure S5). Based on these encouraging results, the diastereomeric mixture of compound 1 was co-crystallized in complex with VCB. “

- An experimental direct comparison of compound 1 to VH101 and a first-generation VHL binder (see: JACS 2012, 134, 4465-4468; Angew. Chem. Int. Ed. 2012, 51, 11463 –11467) via the protein-observed NMR titration protocol would be highly recommended.

We would like to thank the reviewer for this excellent recommendation. For the purpose of this revision, we have conducted the synthesis of example 1 of JACS 2012, 134, 4465-4468, named reference molecule 2 in our study, generated the respective data for comparison (Table 1, Supplementary Figure 5) and revised the discussion.

- NMR experiments: As dose-dependent experiment for compound 1 have been performed via ligand observed displacement assay and protein-observed titrations, the K_i (inhibition constant) value from the displacement assay could be used to validate the K_D value derived from the protein-observed experiments.

As described above, the ^{19}F experiments were performed at a single concentration. Based on the moderate affinity of compound 1, we think that the protein-observed K_D is better suited for the affinity determination.

- K_D determination: Results should stem at least from independent duplicates of the performed assays.

We repeated the titration of compound 1 and reference molecule 2 and have adapted the determined K_D to the average of the two independent experiments as requested.

- The data of the mentioned TR-FRET assay (see lines 207-210) is missing, both in manuscript and SI. Furthermore, the procedure of this assay is missing as well.

We have added the respective data and assay description to the supplementary information.

- The determined IC_{50}/K_D values of the novel compound 1 should be compared to those of established VHL binders. Considering that compound 1 represents an un-optimized first hit, it would be recommended to compare these values not only to VH101 (“VHL binder 10”), but also to early hits from the structure-guided optimization process (see e.g., JACS 2012, 134, 4465-4468; Angew. Chem. Int. Ed. 2012, 51, 11463 –11467). This comparison should also be referred to in the later discussion on the usability of Drugit to identify molecular starting points for ligand optimization.

We again thank the reviewer for the excellent suggestion. As described above, we synthesized example 1 of JACS 2012, 134, 4465-4468 and generated data for comparison (Table 1, Figure S5). The discussion was revised accordingly.

- Table 1 shows in vitro ADME properties. These have not been mentioned at any point in the manuscript, nor in the SI and no protocol/procedure is given for these assays. If these assays have been performed, they should be included in the discussion of the in vitro analysis of compound 1 and details must be added to the SI.

Many thanks for highlighting this, we added details to the supplementary Information and changed text to: *“Due to the provided objective criteria as well as the post-game property filtering, compound 1 is characterized by its improved predicted physicochemical properties, specifically with respect to the lower TPSA (Table 1). Permeability and Caco2 efflux ratio were determined at Boehringer Ingelheim. Compound 1 features a higher intrinsic permeability, and lower Caco-2 efflux ratio compared to reference molecule 1 (Table 1).”*

Co-crystal structure:

- How does the protein structure of the co-crystal of compound 1 with VHL compare with the VHL protein coordinates used in the game – are there significant changes in sidechain conformations? This comparison could be very informative on the impact of keeping the protein atoms fixed in the Drugit framework, and how this might affect the hit-finding process.

Sidechain rotamers of ligand contacting residues are unperturbed, and binding site residues are mostly within 0.5 Ang of the 5NVX structure (and hence the starting structure used in game). The most noticeable change near the binding site is a sidechain rearrangement of His110, though this sidechain does not make any interactions with the ligand, and the rearrangement is likely due to crystal contact differences. Shifts in Arg69 and Arg107 can also be seen, though again these are not making appreciable ligand contacts. The following was added to the text: *“The sidechain and backbone conformations of ligand-contacting residues are not appreciably different from that of the reference molecule 1 bound structure (PDB ID: 5NVX).”*

- As the result of the co-crystal structure with compound 1 in (R,R)- configuration does not match the configuration of the original the player-designed ((R,S)- configuration) it would be very interesting to calculate binding affinity (see lines 183-186) of both conformation for the original player-designed compound and the post-game optimized compound 1 in direct comparison.

The following was added in the main text: *“To allow for a direct comparison, absolute binding free energies for reference molecule 1, the bound diastereomer of compound 1, and the player-designed molecule were calculated. The predictions for compound 1 and the player-designed molecule are within 1 kcal/mol and therefore in the same range, confirming that the key pharmacophores of the player’s design are conserved (Table S7).”* Table S7 was added to the supplement.

Discussion:

- The here presented game designs novel ligands for a protein target based on co-crystal structure of a highly potent ligand of a known binding site, resulting in a >600-fold less potent novel binder (comparing KD values). It would be appreciated if the authors could comment on the productivity/efficiency of this gamified approach of finding novel binders (would experts design better hit candidates from scratch?) and to identify binders for less-explored targets.

It is difficult to directly compare this approach with expert design, as the experiment was not set up as a direct head-to-head comparison. We do note, though, that six of the nineteen compounds selected were created by a practicing medicinal chemist. However, none of the molecules derived from those compounds showed activity, whereas the novel motif was created by someone with no medicinal chemistry experience. Based on reviewer suggestions we included an earlier starting point of the VHL ligand discovery in the manuscript (reference molecule 2), generated the respective data, and revised the discussion accordingly.

- Lines 250-253: The novelty of compound 1 is arguably exchanging Hyp with the hydroxypiperidinone core motif. If the idea is to replace this hydroxypiperidinone core motif in later structure-based ligand optimization, wouldn't structure-based ligand optimization from a more potent Hyp-containing binder (with the goal to replace the Hyp motif) be a more efficient approach?

We apologize, there was a typo in the text. We significantly revised the discussion based on reviewer suggestions.

Referral between figures and text:

In several cases, there is a disconnect between the description of figure subsections in the main text and the content of this subsections, such as

- Figure 1a depicts the atom selection tool of the Small-Molecule Design panel of the Drugit interface, but in the main text Figure 1a is describe as the main window showing the "3D structure of a starting molecule docked" (lines 76-78). This view is in fact presented in Figure 1c.
- Figure 1d shows the undo function of Drugit but is referred to as Figures 1c in the main text (line 110).

The references have been corrected accordingly.

- The description of figure 2b in the main text (lines 156-158) is misleading, as all 6,500 compounds docked to VHL are shown here, not the reduced library of 19 compounds.

A sentence was added to clarify this point. *"Figure 2b shows the 1,073 remaining player suggestions within the VHL binding site, while Figure 2c compares the chemical space covered by player suggestions vs. known VHL binders used as spike molecules during property filtering."*

- There is no mention of Figure 2c in the main text. A discussion of the chemical space of in-game designed compounds vs current VHL inhibitors would be highly desirable. In the current version, Figure 2c does not seem to have any purpose. How has the PCA been performed? There is no description of the PCA in either the manuscript or the SI.

We have clarified this point in the main text as follows: *"Figure 2b shows the 1,073 remaining player suggestions within the VHL binding site, while Figure 2c compares the chemical space covered by player suggestions vs. known VHL binders used as spike molecules during property filtering. Interestingly, the player-designed molecules (black dots) cover a wider chemical space as defined by the two main principal component axes than the known VHL binders (black dots with blue circles)." A short explanation on the PCA has been added to the Figure legend of Figure 2c: "RDkit descriptor calculations and a PCA were carried out with default settings in MOE 2022.02 (Molecular Operating Environment, Chemical Computing Group), including a normalization of descriptors."*

Figures & Tables:

- Table 1 lacks a description. References must be included for experimental values of "small molecule 10" taken from earlier reports.

A title for Table 1 has been added. A reference for the FP assay for small molecule 10 (named reference molecule 1 now) has been added to the Table footnotes.

- Figure 3a: The colors in the 1D spectrum of (a) are not clearly identifiable. I would recommend increasing the line width of the signals for better distinguishability.

The figure was revised according to the reviewer's suggestion and the line width was increased.

- Figures 3: In the description of (a), the color for the signal after addition of compound 1 should be included.

We added the respective information.

- Figure 4(a,b): As the discussion refers to potential optimization of compound 1 by engaging with Trp88 and Phe91 via π - π interactions, it would be highly recommended to select a view of the binding site of VHL that includes both residues. Currently, Phe91 is not shown.

We agree with the reviewer and included Phe91 in the figures 2a and 4a,b,c.

- Figure 4b: is the protein structure derived from PDB: 8POF (VHL w/ 1) or 5NVX (VHL w/ "compound 10" (VH101))?

Many thanks for highlighting this lack of clarity. Protein atoms in figure 4 a, b and c originate from PDB: 8POF and the legend has been complemented.

Supporting Information:

- Table S1: "no center: small molecule 10 lacking the hydroxyproline sidechain": Does that mean replacement of Hyp with glycine? If so, please state it more clearly.

Table S2 has been added, showing the structures of the starting ligands given to players.

- Please add protocols/procedures for the ^{19}F displacement assay (compound screen as well as dose-dependent assay), TR-FRET assay, determination of aqueous solubility and Caco-2 permeability, expression/labelling of ^{13}C methyl labelled (Ile, Leu, Val) VCB complex.

The procedures and references for the respective methods have been added to the supplementary information.

- Add information (table) on %displacement of all tested compounds of the initial NMR displacement assay.

The respective data has been added in Supplementary Table 6.

- The section on general synthetic methods, materials (e.g., sources for commercial compounds) and instrumentation is missing. Please check the author guidelines in the methods section on required information.
- References for already known compounds are missing.
- Yields in mmol, R_f values and melting points missing in the compound characterization data.
- LC-MS results: Calculated m/z of the [M+H]⁺ must be added in the compound characterization data. Is the accuracy of the used instrument really m/z = ± 1, viz. are all significant decimals shown?

- Characterization of chiral compounds is completely missing. This includes Details on chiral SFC purification in the synthesis section (e.g., for compounds 20 and 21), determination of $[\alpha]_D$ values, characterization using appropriate techniques, including polarimetry, NMR, single crystal XRD, or by correlation of HPLC or GC (see characterization of chiral compounds section in author guidelines).

We thank the reviewer for the guidance to improve the supplementary materials. We have obtained and reported the required information to characterize the compounds. With respect to m/z values in LC-MS measurement, all significant decimals are shown.

- Information on purity of compounds is missing. Purity cannot be checked as NMR spectra aren't attached.

We have attached all ^1H and ^{13}C NMR spectra.

Further small corrections:

- Line 31: "designed by a player": This wording in the abstract seems a bit misleading – while the core hydroxy-piperidinone structure has been designed by a player, its eastern and western extension have been substantially modified in post-game processing.

We agree and have removed the statement.

- I would recommend referring to "small molecule 10" as VH101, which is the more known (and less ambiguous) term for this molecule. Regardless of this, the naming of this compound should be consistent throughout the manuscript – currently it is referred to as "small molecule 10", "VHL binder 10" and "ligand 10". It might be beneficial to add the chemical structure of VH101 to Figure 2a.

"Compound 10" from Soares et al. (2018) J Med Chem, crystalized in PDB ID 5NVX, is chemically distinct from VH101. We recognized that the naming creates confusion and decided to name molecules which have been described previously as reference molecules 1-2 and include them in Figure 4.

- I would recommend to either use "western" and "eastern" or "left-hand side" and "right-hand side" to refer to different parts of the small molecule, instead of mixing these terms.

We have updated the paper to use the terminology and abbreviations introduced in J. Med. Chem. 2014, 57, 8657, namely left-hand side (LHS) and right-hand side (RHS) ([dx.doi.org/10.1021/jm5011258](https://doi.org/10.1021/jm5011258) |).

- I would recommend using consistent nomenclature to refer to "larger than" – either "260+" or ">260".

Terminology has been normalized to >260.

Reviewer #2 (Remarks to the Author):

The authors describe the development of "Drugit", a small molecule design mode of popular citizens science game Foldit. The authors then test the ability of users to design a binder to VHL, a popular ligase in the TPD field and after extensive filtering of designs arrive at a 250 uM binder designed by a

non-medicinal chemist user.

This is a clearly written and concise story on how citizens science can help in guiding the development of new molecules for challenging and yet interesting targets such as VHL.

I would recommend publishing this report in its current form. My only suggestion is to add a link to the website where users can play Drugit.

We thank the reviewer for their kind words. Links to <https://drugit.org> have been added to the text.

Reviewer #3 (Remarks to the Author):

This is quite an original paper investigating whether “citizen scientists”, by using an on-line simple tool (“Foldit”), manage to identify drug leads. One of the molecules identified by a pool of people with no background in medicinal chemistry turned out to bind to the chosen protein target (a ligase); the subsequent X-ray structure showed that a binding mode like that predicted by the citizen scientist. The approach suggests that citizen scientists could be involved in the highly costly process of structure-based drug design. This work is of high significance in the field of computer-aided drug design. It uses a sound methodology, and it can be easily reproduced (especially the on-line applications). The paper is nice to read and very well organized.

Minor points:

- The societal impact of the work could be added.

The following was added to the discussion section: *“By providing small molecule design tools to anyone who can download and install a game onto their computer, we expand the range of people who contribute to drug design. Democratizing the process in this way can bring extra resources to discovering drugs for rare and neglected diseases.”*

- The limitations of the on-line docking method relative to the state-of-the-art could be summarized.

We acknowledge that the current in-game evaluation metrics are less than ideal. Incorporating more state-of-the-art methods could indeed help, however we do not as-of-yet have a head-to-head comparison. (Nor does this work lend itself to such.) The following was added to the discussion section *“Further improvement of binding evaluation in-client, potentially by incorporating other physics and machine learning based scoring, should increase the rate of player success, as would investigating orthogonal approaches for post-game compound evaluation. “*

- The knowledge required to run the on-line applications should be stated.

The goal of the game is to provide small molecule design tools to members of the public, without specialized knowledge of drug development. To get started, one only needs to be able to download and install the game client. The game client includes some preliminary tutorials for small molecule design. How best to guide citizen scientists once they have installed the game is a topic of ongoing research. The following has been added to the manuscript: *“the client is intended to be usable for*

people with all levels of medicinal and biochemical experience, and includes basic tutorials on how to use the provided tools"

- Enzymes are relatively easy targets: could this approach be used, for instance, to allosteric ligands of GPCRs or ion channels?

As a subdomain of a larger E3 ligase complex, VHL is involved in protein-protein recognition *in vivo* (the cleft being targeted is not an enzyme active site). We don't anticipate any issues with other targets, so long as there's a sufficient pocket for a small molecule ligand to bind. We are currently testing the approach with other proteins, including GPCRs.

Reviewer #4 (Remarks to the Author):

One of the most important problems in the application of molecular biophysics to medicine is drug design. The authors address early stages of the process by crowdsourcing through a game, Drugit, inspired by and building on Foldit, a broadly known application of Rosetta energy function to protein folding and design. This study presents a single test case based on VHL E3 ligase: from posing the task to experimental validation of the designed molecule by X-ray crystallography, among other methods, thus demonstrating the success of the design. Although promising, it is important to remember that this is an early (and possibly the simplest) stage of drug design, and 90% of the road is still ahead (which may be good to mention in the manuscript).

A more general question is whether Rosetta energy function is a good indicator of success in drug design. The authors indicate that it might not be. What are possible alternatives?

This is an active area of research for us, and it is still too early to give a good answer. Other physics-based or machine learning scoring approaches may give better ranking of compounds, especially if combined in a "consensus" approach. Information from the CACHE challenge (in which we are participating) may help to provide further information on how to best evaluate compounds. To provide better insights to the reader regarding future directions, we altered a sentence to read *"Further improvement of binding evaluation in-client, potentially by incorporating other physics- and machine learning-based scoring, should increase the rate of player success."*

Using "objectives" is a nice way to address the usual problems in drug design that are not related to binding energy.

Overall, this work presents a detailed description of the experiment and should be interesting to the readers. Although I remain a bit skeptical whether the Drugit module holds a promising future in drug design, the experiment presented in the paper is well-designed and well-presented. It is difficult to suggest any improvements to this work as it stands, without significant reconsideration.

Just out of curiosity, the protein selected for the experiment is rather small, and probably smaller than a typical drug target. Any issues with the Drugit game/players to work with larger targets?

There are some performance considerations on older machines with larger systems. However, we have recently been experimenting with larger systems, and with the appropriate puzzle setup the

size limitations do not seem to be as much of a restriction as we initially feared. We have been able to successfully run puzzles with larger proteins (such as full GPCR transmembrane domains).

Small matters:

1. Consistency in punctuation: 'Drugit' (quotes of different style).

Typography has been corrected.

2. Maybe it is better to use "dimension 1" and "2" instead of "0" and "1" in Fig. 2? These are known as the first and the second principal components. There is no 0th component. You already use "compound 1" and not "compound 0".

We corrected the legend accordingly.

3. Fig. 2 might be better to use "player-designed" with a hyphen.

Hyphenation has been normalized.